# HYPERBOLIC SPACE PATH AGGREGATION FOR OVER-SQUASHING IN MULTI-RELATIONAL GRAPH

## ABSTRACT

The link prediction task has attracted significant attention from the graph communities. However, GNN-based methods still exhibit subpar performance in the link prediction task for large-scale, multi-relational knowledge graphs. Previous works utilize hyperbolic space to model hierarchical relations and employ path aggregation to alleviate the over-smoothing problem. The two approaches are complementary in their advantages, but both encounter the issue of over-squashing. The former experiences curvature collapse during training, while the latter struggles to distinguish the similar entities connected by the same relation. Specifically, we utilize hyperbolic space path aggregation for the curvature stability and anti-symmetry weight in the update process to alleviate the issue of over-squashing. Our method achieves improvements on two standard transductive datasets and eight inductive versions. Further analysis reveals the potential relationship between curvature and types of relation.

## 1 INTRODUCTION

Knowledge graph contains a large number of structured facts $(h, r, t)$, where a fact expresses a directed relation $r$ from a head entity $h$ to a tail entity $t$. The complex KGs, such as FreeBase (Berant et al., 2013), DBPedia (Auer et al., 2007), and Wikidata (Vrandečić & Krötzsch, 2014), are manual or automatic collections from structured or unstructured heterogeneous data on the web. Knowledge Graph has supported many downstream applications, including question answering (Berant et al., 2013), recommender systems (Vrandečić & Krötzsch, 2014), and graph retrieval-augmented generation (Edge et al., 2024). However, knowledge graphs cannot encompass all the rich facts of the open world. Therefore, the multi-relational link prediction task has received widespread attention in the research community.

Multi-relational link prediction aims to infer missing facts in knowledge graph triplets and enhance the structural integrity of knowledge graphs. Previous methods can be broadly categorized into several paradigms. Heuristic embedding-based methods (Bordes et al., 2013; Yang et al., 2015; Sun et al.; Li et al., 2022a; Xiao & Cao, 2024) employ heuristic relational operations to embed triplets into the low-dimension semantic space, enabling the model to capture relational patterns such as symmetry, anti-symmetry, inversion, and composition. Graph neural network-based methods (Vashishth et al., 2020; Zhu et al., 2021; Zhang & Yao, 2022; Zhu et al., 2023; Zhang et al., 2023) capture relational local interactions through iterative neighborhood aggregation, with advantages in being lightweight and providing path explanations. Relational attention-based methods (Shang et al., 2024; Chen et al., 2021; Liu et al., 2024) use relational attention or transformer to capture global information, which enhances the model capacity for the noisy links; however, it often results in a larger number of parameters. Large language model-based methods (Wang et al., 2022; Saxena et al., 2022; Liu et al., 2022) align the semantic information from vast amounts of text with the structural features within knowledge graphs, achieving excellent integration results for certain richly described datasets (such as WN18RR); however, their assistance is limited for private and complex data. Hyperbolic space-based methods (Montella et al., 2021; Balaževic et al., 2019; Montella et al., 2021; Liang et al., 2024a) attempt to model the non-uniformity of graphs through non-Euclidean geometries with different curvatures, proving particularly effective for hierarchical relations.

Recent works, particularly those based on path-based graph neural networks and hyperbolic space with negative curvature, have provided striking insights. First, path-based work transforms indepen-

dent node and relation features into path features conditioned on the query relation. Incorporating path-ordering in local structure aggregation can enhance the model's ability to represent frequently occurring relational patterns and address the feature collapse caused by the over-smoothing problem (Zhu et al., 2021). Second, work based on hyperbolic space leverages the property of exponential expansion in hyperbolic space, which is beneficial for hierarchical relations (Liang et al., 2024b) (tending towards tree-like structures).

However, graph neural network in hyperbolic space faces the issue of curvature collapse, making stable multi-layer training difficult. Previous works based on hyperbolic space have employed heuristic score functions (such as MuPE (Balaževic et al., 2019) and AttH (Montella et al., 2021)) or shallow layers (HGCN (Chami et al., 2019) utilizing only two layers), which limit the true expressive power of hyperbolic space. Through further analysis of HGCN in appendix E, we found that the greater number of layers in HGCN leads to a gradual decline in performance across most datasets, as well as a faster curvature collapse. These phenomena indicate that existing modeling methods based on hyperbolic space encounter issues of over-smoothing (similar node features after multi-aggregation) and over-squashing (feature compression on long-range or aggregated structures).

Previous works based on path aggregation have low performance on *one-to-many* tails and *many-to-one* heads. We provide *Balanced Forman curvature* (Topping et al., 2022) and the MRR Metric about different relation categories in the WN18RR and FB15k237. E Through these analyses, we can understand that the phenomenon of over-squashing is common in knowledge graphs.

We propose a model, namely HypPath, that gains advantages by path aggregation in hyperbolic space. Specifically, we enhanced the expressive power of path aggregation features by utilizing the Möbius operator and distances in hyperbolic space. Furthermore, we utilized anti-symmetric weight representing the rotational invariance of hyperbolic space and tangent space at the origin to update path features, alleviating the over-squashing. Finally, we combine Euclidean score and Hyperbolic distances to form the final scores of triplets.

Our main contributions are as follows,

- We are the first to express KGs as path aggregation in the hyperbolic space and propose a typical model, namely HyPNet.
- We define an update function in Hyperbolic space to alleviate over-squashing in KGs and theoretically analyze the convergence of the module defined in hyperbolic space.
- We achieve improvement with fewer parameters on two standard transductive datasets, WN18RR and FB15k-37, and eight inductive versions.

## 2 RELATED WORK

### 2.1 MULTI-RELATIONAL LINK PREDICTION

**Heuristic Embedding Methods** Early research about multi-relational link prediction primarily operated in Euclidean space, designing explicit score functions to evaluate the interaction of triplets. TransE (Bordes et al., 2013) is a pioneering work in this direction, modeling relations as translation vectors between entity embeddings. Subsequent improvements focused on the expressive score function. DistMult (Yang et al., 2015) uses diagonal matrices to capture symmetric relations; RotatE (Sun et al.) encodes relations as rotations in complex Euclidean space to model anti-symmetry and inverse relations; House (Li et al., 2022a) leverages Householder parameterization to enhance the expressiveness of embedding transformations; TuckER (Balazevic et al., 2019) captures high-order interactions between entities and relations through powerful tensor decomposition. Recent advancements, such as MIG-TF (Yusupov et al., 2025) and HAQE (Liang et al., 2024b), further improve the modeling of compositional and symmetric relations. Although this paradigm is scalable and computationally efficient, it often neglects the global topology of the graph (e.g., multi-hop paths), limiting its ability to capture complex structural dependencies.

**Graph Neural Network-Based Methods** GNN-based methods encode the topological structure of the graph to model multi-hop relational dependencies, compensating for the structural neglect of Euclidean heuristic methods. Early works such as CompGCN (Vashishth et al., 2020), adapted composition graph convolutional networks to knowledge graphs. Neural Bellman-Ford Networks

(NBFNet) (Zhu et al., 2021) established a general path-based reasoning framework. Recent methods optimize the balance between local and global information through adaptive message-passing mechanisms, such as Adaprop (Zhang et al., 2023) and A* Net (Zhu et al., 2023). In contrast, MGTCA (Shang et al., 2024) integrates Euclidean and hyperbolic spaces in convolutional attention to model heterogeneous relational patterns. This paradigm still faces limitations, including over-squashing in deep architectures and a limited ability to capture long-range dependencies. Logic rule-based methods incorporate symbolic reasoning into embedding learning to improve interpretability and generalization in KGC. DRUM (Sadeghian et al., 2019) and RNNLogic (Qu et al., 2020) learn differentiable logical rules via neural networks, while RulE (Tang et al., 2024) jointly embeds entities, relations, and logic rules for end-to-end reasoning. This paradigm offers interpretability advantages, but its scalability is limited on large knowledge graphs, and it may struggle to generalize to rare or unseen relations.

**Large Language Model-Based Methods** Pretrained large language models (LLMs) introduce a new paradigm for KGC, leveraging semantic information from entity textual descriptions for global reasoning. SimKGC (Wang et al., 2022) demonstrates that strong textual representations alone can achieve competitive performance. KGT5 (Saxena et al., 2022) directly generates missing triplets, while transformer-based frameworks such as KnowFormer (Liu et al., 2024) integrate textual information with knowledge graph structure, achieving strong performance. However, this paradigm heavily relies on high-quality textual data and may lose fine-grained structural information when converting graphs into sequences.

## 2.2 Hyperbolic Geometry

Hyperbolic geometry has emerged as a particularly powerful paradigm due to its superior ability to represent hierarchical structures. Compared to Euclidean approaches, hyperbolic methods show significant advantages in embedding hierarchical data. MuRP (Balažević et al., 2019) proposes multi-relational Poincaré embeddings, extending the Poincaré ball model to multi-relational knowledge graphs. HGCN (Chami et al., 2019) combines message passing with hyperbolic geometry to efficiently aggregate information along hierarchical paths. Recent extensions include temporal hyperbolic embeddings introducing relation and time curvature ATTH (Montella et al., 2021), and FHRE (Liang et al., 2024a) that perform all transformations (e.g., rotations) entirely in hyperbolic space to preserve complex relational structures.

## 3 Background and preliminaries

### 3.1 Multi-Relational Link Prediction

*Knowledge Graph* consists of a set of triplets $\{(h_i, r_j, t_k)\} \subseteq \mathcal{E} \times \mathcal{R} \times \mathcal{E}$, where $\mathcal{E}$ is a finite entity set and $\mathcal{R}$ is a finite relation set. Each fact $(h_i, r_j, t_k)$ respectively denotes a directed relation $r_j$ from the head entity $h_i$ to tail entity $t_k$. Although knowledge graphs contain large numbers of facts, they are still incomplete due to the complex nature of the real world. Multi-relational link prediction aims to predict the other potential facts based on existing facts. In that way, *Score Function* $\hat{\mathcal{X}}$ is an approximation of $\mathcal{X} = \{0, 1\}^{|\mathcal{E}| \times |\mathcal{R}| \times |\mathcal{E}|}$ which denotes a third-order binary tensor to indicate whether each fact is correct or not. $|\mathcal{E}|$ and $|\mathcal{R}|$ denote the number of entities and relations. The problem is reduced to ranking a set of candidates and selecting the most likely entity that makes the query $(h, r_q, ?)$ correct. There are many types of score functions, such as the translational model and the graph neural network. We use a path-based approach to model the score function as the paths aggregation between two points conditioned on the query $\hat{\mathcal{X}}_{ijk}(t_k|h_i, r_j)$.

### 3.2 Hyperbolic Geometry

Hyperbolic space is a geometric structure characterized by *constant negative curvature* $-c(c > 0)$, whose spatial form exhibits exponential expansion as the radius increases, in contrast to the uniform linear growth of Euclidean space. This characteristic makes hyperbolic space suitable for modeling hierarchical relationships, enhancing the expressive power for locally uneven knowledge graphs, such as queries associated with the same entity or entities associated with the same query.

Commonly used hyperbolic models include the hyperboloid model and the poincaré ball model. There is an isomorphic mapping (one-to-one, distance-preserving) between them.

**Poincaré Ball Model** The poincaré ball $\mathbb{B}_c^d = \left\{ \boldsymbol{x} \in \mathbb{R}^d : c\|\boldsymbol{x}\| < 1 \right\}$ is analogous to an $n$-dimensional manifold centered at the origin point $\mathbf{0}$ with a radius of $c$. The essence is to represent the non-Euclidean hyperbolic geometric structure using spheres in Euclidean space. The tangent space $\mathcal{T}_{\boldsymbol{x}}\mathbb{B}_c^d = \mathbb{R}^d$ is the linear expansion at point $\boldsymbol{x}$, and all linear transformations between tensors are operated within the tangent space. The addition, scalar multiplication, and matrix-vector multiplication defined in Euclidean space can be generalized to hyperbolic space. Because of the *Rotational invariance* at origin $\mathbf{0}$, we only utilize the tangent space at the origin $\mathbf{0}$ and the three Möbius operations mentioned in the following equations. More details about any point $\boldsymbol{x}$ are provided in the appendix D. $\| \cdot \|$ denotes the Euclidean norm and $< \cdot, \cdot >$ denotes the Euclidean inner product.

Given $\boldsymbol{x} \in \mathbb{B}_c^d$ and $\boldsymbol{y} \in \mathcal{T}_{\boldsymbol{x}}\mathbb{B}_c^d$, *exponential map* $\log_{\mathbf{0}}^c : \mathcal{T}_{\boldsymbol{x}}\mathbb{B}_c^d \to \mathbb{B}_c^d$ and *logarithmic map* $\exp_{\mathbf{0}}^c : \mathbb{B}_c^d \to \mathcal{T}_{\boldsymbol{x}}\mathbb{B}_c^d$ are two inverse mappings between poincaré ball $\mathbb{B}_c^d$ and its tangent space $\mathcal{T}_{\boldsymbol{x}}\mathbb{B}_c^d$.

$$\exp_{\mathbf{0}}^c(\boldsymbol{x}, c) = \tanh(\sqrt{c}\|\boldsymbol{x}\|)\frac{\boldsymbol{x}}{\sqrt{c}\|\boldsymbol{x}\|}, \quad \log_{\mathbf{0}}^c(\boldsymbol{y}, c) = \operatorname{arctanh}(\sqrt{c}\|\boldsymbol{y}\|)\frac{\boldsymbol{y}}{\sqrt{c}\|\boldsymbol{y}\|}. \tag{1}$$

- *Möbius addition* $\oplus_c$ can ensure that two elements $\boldsymbol{x}, \boldsymbol{y} \in \mathbb{B}_c^d$, the results remain closed within hyperbolic space, adapting to hierarchical complex structures.

$$\boldsymbol{x} \oplus_c \boldsymbol{y} = \frac{(1 + 2c\langle\boldsymbol{x}, \boldsymbol{y}\rangle + c\|\boldsymbol{y}\|^2)\boldsymbol{x} + (1 - c\|\boldsymbol{x}\|^2)\boldsymbol{y}}{1 + 2c\langle\boldsymbol{x}, \boldsymbol{y}\rangle + c^2\|\boldsymbol{x}\|^2\|\boldsymbol{y}\|^2}. \tag{2}$$

- *Möbius scalar multiplication* $\otimes_c$ scales a vector $\boldsymbol{x} \in \mathbb{B}_c^d$ by the real number $b$ without changing its direction, and restricts the result within hyperbolic space.

$$b \otimes_c \boldsymbol{x} = \tanh(b \operatorname{arctanh}(\sqrt{c}\|\boldsymbol{x}\|))\frac{x}{\sqrt{c}\|\boldsymbol{x}\|}. \tag{3}$$

- *Möbius matrix-vector multiplication* $\otimes_c$ apply a linear transformations $\boldsymbol{M} \in \mathcal{M}_{m,n}\{\mathbb{R}\}$ to the vector $\boldsymbol{x} \in \mathbb{B}_c^d$. Rotation invariance means $\boldsymbol{M} \otimes_c \boldsymbol{x} = \boldsymbol{M}\boldsymbol{x}$ for all $\boldsymbol{M} \in \mathcal{O}_d(\mathbb{R})$ the rotation around the origin $\mathbf{0}$ is consistent for the tangent space $\mathcal{T}_{\boldsymbol{x}}\mathbb{B}_c^d$ and the hyperbolic space $\mathbb{B}_c^d$.

$$\boldsymbol{M} \otimes_c \boldsymbol{x} = \tanh(\frac{\|\boldsymbol{M}\boldsymbol{x}\|}{\|\boldsymbol{x}\|}\operatorname{arctanh}(\sqrt{c}\|\boldsymbol{x}\|))\frac{\boldsymbol{M}\boldsymbol{x}}{\sqrt{c}\|\boldsymbol{M}\boldsymbol{x}\|}. \tag{4}$$

- Apart from the Möbius addition, the other Möbius operations mentioned above satisfy the general formula as follows. Notice that, if $f$ is a point-wise non-linearity, such as layer normalization or activation functions, the following equation still holds. Given $\boldsymbol{x}^H \in \mathbb{B}_c^d$,

$$f^{\otimes_c}(\boldsymbol{x}^H) = \exp_{\mathbf{0}}^c(f(\log_{\mathbf{0}}^c(\boldsymbol{x}^H))). \tag{5}$$

For the *composition function* of several operations, the intermediate exponential map and logarithmic map can be canceled out $f_k^{\otimes_c} \circ \cdots \circ f_1^{\otimes_c} = \exp_{\mathbf{0}}^c \circ f_k \circ \cdots \circ f_1 \circ \log_{\mathbf{0}}^c$. Therefore, we only need to perform a map between hyperbolic and tangent spaces once when encountering addition or matrix-vector multiplication.

- *Geodesic distance* is the shortest path between two points in hyperbolic space $\boldsymbol{x}, \boldsymbol{y} \in \mathbb{B}_c^d$,

$$d_c^{\mathbb{B}}(\boldsymbol{x}, \boldsymbol{y}) = \frac{2}{\sqrt{c}}\tanh^{-1}(\sqrt{c}\| - \boldsymbol{x} \oplus_c \boldsymbol{y}\|). \tag{6}$$

### 3.3 PATH AGGREGATION

Graph neural networks develop rapidly on multi-relational graphs, but suffer from issues such as heterogeneity, over-smoothing, and over-squashing. GNN has developed various approaches to link prediction, including vanilla, composition, spectral, rewiring, subgraph, and path methods. The representation feature conditioned on the query and path is more expressive and effective than vanilla message passing. Since the path aggregation conditioned on the query alleviates the over-smoothing problem, path-based methods demonstrate superior performance and versatility. Given a query

$(h, q, ?)$ over a graph $\mathcal{G}$, the *Neural Bellman-Ford Network* (Zhu et al., 2021) expresses the path aggregation as a composition of single-step message passing, where $n$ denotes the layers, $d$ denotes the dimensionality, $\boldsymbol{Q}$ denotes the initial query embedding matrix, and $*$ denotes element-wise multiplication,

$$\boldsymbol{z}_q^0(h, t) = \text{INDICATOR}(h, t, r_q) = \mathbf{1}_{u=v} * \boldsymbol{Q}, \quad v \in \mathcal{E} \tag{7}$$

$$\boldsymbol{z}_q^n(h, t) = \text{UPDATE}\Big(\boldsymbol{z}_q^{n-1}(h, t), \text{AGGREGATE}\big(\text{MESSAGE}(\boldsymbol{z}_q^{n-1}(h, x), \boldsymbol{e}_r(x, t)) | x \in \mathcal{E}(h), r \in \mathcal{R}\big)\Big).$$

The path-based approach can adapt to both transductive and inductive experimental settings due to generalization for unseen entities. Finding a way to achieve a deep, stable, and expressive conditioned message passing is key to addressing large-scale and sparse multi-relational graphs.

## 4 METHODOLOGY

### 4.1 FRAMEWORK

In the following section, we emphasize the poincaré ball model and relational path aggregation as the foundation of our work. We reconstructed the aforementioned Indicator, Message, and Update functions using operators in hyperbolic space.

**Hyperbolic Space Path Aggregation**

Given a query $(h, q, ?)$ over a graph $\mathcal{G}$, we define the curvatures $c_q$ of the poincaré ball conditioned on the query relation $q$, and initial edge representation for each encoder layer.

The *indicator function* determines the initial representation of the path, which is dictated by the query relation. We initialize the query embedding $\boldsymbol{q} \in \mathbb{R}^d$, and use the exponential map to map the query embedding in Euclidean space to hyperbolic space $\boldsymbol{z}_q^0 = \boldsymbol{q}$. Unless otherwise specified, all vectors mentioned in the following text are defined in Euclidean space to simplify understanding.

The *message function* computes the $n$-hop path feature using the $(n\text{-}1)$-hop path features and the current edge feature $\boldsymbol{e}_q(x, t)$. Due to the particularity of addition in hyperbolic space and the properties of composition functions in the tangent space, we directly utilize multiplication in Euclidean space rather than addition, that is MESSAGE : $\boldsymbol{m}_q^n = \boldsymbol{z}_q^{n-1}(h, x) * \boldsymbol{e}_r^n(x, t)$. Our edge features are designed with flexibility to handle structures of various complexities. An independent version is designed for the relatively simpler structure, such as on WN18RR. And a dependent version is designed for the more complex structure, such as on FB15k-237,

$$\begin{aligned}
\boldsymbol{e}_r &= \log_{\boldsymbol{0}}^{c_q}\big(-\exp_{\boldsymbol{0}}^{c_q}(\boldsymbol{q}^0) \oplus_c \exp_{\boldsymbol{0}}^{c_q}(\boldsymbol{b}_r^n)\big), \\
\boldsymbol{e}_r &= \log_{\boldsymbol{0}}^{c_q}\big(\boldsymbol{W}_r^n \otimes_c \exp_{\boldsymbol{0}}^{c_q}(\boldsymbol{q}^0) \oplus_c \exp_{\boldsymbol{0}}^{c_q}(\boldsymbol{b}_r^n)\big).
\end{aligned} \tag{8}$$

The *aggregation function* is a set operator that is independent of the path set. We still use principal neighborhood aggregation(PNA) (Corso et al., 2020) as the NBFNet does, which is a combination of maximum, minimum, average, and summation. Finally, the aggregated path features are obtained through a linear transformation $g_1(\cdot) : \mathbb{R}^{13d} \to \mathbb{R}^d$ for dimensionality reduction $\Phi(\boldsymbol{M}_q^n) = g_1(PNA(\boldsymbol{m}_q^n(x))), x \in \mathcal{E}(h)$.

The *update function* is particularly noteworthy; we design a novel update mechanism to enhance the training stability of deeper models. We define an anti-symmetric weight $\boldsymbol{A} = \boldsymbol{M} - \boldsymbol{M}^T - \gamma\boldsymbol{I}$ and bias $\boldsymbol{b}$ to transform the path feature, where $\gamma$ is a scale factor. Notice that we use the same anti-symmetric and bias for each encoder layer. For simplicity, we omit the Layer Normalization (Ba et al., 2016) and the Activation Function ReLU preceding the logarithmic map in the update function.

$$\text{UPDATE} : \boldsymbol{z}_q^n = \boldsymbol{z}_q^{n-1} + \sigma\big(\log_{\boldsymbol{0}}^{c_q}\big(\exp_{\boldsymbol{0}}^{c_q}(\boldsymbol{A}\boldsymbol{z}_q^{n-1}) \oplus \exp_{\boldsymbol{0}}^{c_q}(\Phi(\boldsymbol{M}_q^n(x))) \oplus \exp_{\boldsymbol{0}}^{c_q}(\boldsymbol{b}))\big). \tag{9}$$

**Theorem 1.** *For the update function defined in Euclidean space,*

$$\boldsymbol{z}_q^n = \boldsymbol{z}_q^{n-1} + \sigma(\boldsymbol{A}\boldsymbol{z}_q^{n-1} + \Phi(\boldsymbol{M}_q(x)) + \boldsymbol{b}), \quad \boldsymbol{x} \in \mathcal{E}(h), t \notin \mathcal{E}(h). \tag{10}$$

*If $\boldsymbol{A}$ is an anti-symmetric matrix, the solution $\tilde{\boldsymbol{z}}_q^n$ of the update function Eq. (10) converges. Furthermore, the update function defined in the Hyperbolic space Eq. (9) also satisfies. The implementation using Euclidean operators is an upper bound for the implementation using hyperbolic operators.*

**Proof 1.** *See in the appendix C.*

The learnable curvature and the model parameters of HGCN often experience asynchronous learning processes, which can lead to curvature collapse when stacking high layer. Anti-symmetric matrices $A$ can maintain the norm of $z_q^n$ constant across different layers $n$, preserving its representation without shrinkage during the aggregation process. The training stability allows the stacking of more layers in GNNs. Experimentally, HyPNet stabilizes the learnable curvature, ultimately improving the model's performance.

The *score function* consists of two components, the main component use a simple two layer MLP with ReLU $g_2(\cdot) : \mathbb{R}^d \to \mathbb{R}^1$ as decoder, while the additional term is the hyperbolic distance in the poincaré ball, i.e.,

$$s(h,q,t) = -d_c^{\mathbb{B}}(\mathbf{0}, \exp_{\mathbf{0}}^{c_q}(z_q^n(h,t))) + g_2(z_q^n(h,t)). \tag{11}$$

Final logits $p(h,q,t) = \sigma(s(h,q,t))$ denote the score of a node $v$ to be a tail of the initial query $(h,q,?)$. Any inductive link prediction model only requires deterministic relation features to adapt to the inductive setting. HyPNet can be trained on any entity with a deterministic relation set, enhancing expressiveness and scalability.

The *loss function* is binary cross entropy over positive and sampled negative triplets, which is a standard practice for multi-relational link prediction,

$$\mathcal{L} = -\log p(h,q,t) - \sum_{i=1}^{n} \frac{1}{n} \log(1 - p(h_i', q, t_i')), \tag{12}$$

where $(h,q,t)$ is a positive triple in the graph and $\{(h_i', q, t_i')\}_{i=1}^{n}$ are negative samples obtained by corrupting either the head $u$ or tail $v$ of the positive sample.

## 5 EXPERIMENT

### 5.1 DATESETS

We use two standard transductive datasets and eight inductive versions extracted from FB15k-237 and WN18RR. Detailed statistics of the dataset can be found in the appendix D. We use reciprocal facts $(t, r^{-1}, h)$ for data augmentation and message passing. $r^{-1}$ is the inverse relation of $r$. Each inverse relation is treated as an independent new relation.

### 5.2 BASELINES

We compare HyPNet with several baselines. The transductive Multi-relational Link Prediction task includes 20 baseline methods with five different types, including Euclidean space-based methods TransE (Bordes et al., 2013), DistMult (Yang et al., 2015), RotatE (Sun et al.), TuckER (Balazevic et al., 2019), RulE (Tang et al., 2024), and MetaSD (Li et al., 2022b); large language model-based methods KGT5 (Saxena et al., 2022), SimKGC (Wang et al., 2022), N-Former (Liu et al., 2022); graph neural network-based methods CompGCN (Vashishth et al., 2020) RED-GNN (Zhang & Yao, 2022), AdaProp (Zhang et al., 2023), ULTRA (Galkin et al.), NBFNet (Zhu et al., 2021); relational attention-based methods HittER (Chen et al., 2021), MGTCA (Shang et al., 2024), KnowFormer (Liu et al., 2024); Hyperbolic space-based methods MuRP (Balaževic et al., 2019), ATTH (Montella et al., 2021), FHRE (Liang et al., 2024a). The Inductive Multi-relational Link Prediction task includes 7 baseline methods, including DRUM (Sadeghian et al., 2019), RED-GNN (Zhang & Yao, 2022), A*Net (Zhu et al., 2023), AdaProp (Zhang et al., 2023), Ingram (Lee et al., 2023), SimKGC (Wang et al., 2022), NBFNet (Zhu et al., 2021), and KnowFormer (Liu et al., 2024).

### 5.3 TRANSDUCTIVE PERFORMANCE

In the transductive multi-relational link Prediction task, we selected two standard datasets, FB15k-237 (Toutanova et al., 2015) and WN18RR (Dettmers et al., 2018), to evaluate the performance of HyPNet. As shown in Table 1, HyPNet surpasses two types of foundation models: path-based methods and hyperbolic space-based methods, demonstrating that the advantages of both can be complementary to some extent.

Table 1: Transductive Multi-relational Link Prediction Results. Best results are bold, second results are underlined.

| Class | Method | FB15k-237 | | | | WN18RR | | | |
|---|---|---|---|---|---|---|---|---|---|
| | | MRR | H@1 | H@3 | H@10 | MRR | H@1 | H@3 | H@10 |
| **Euclidean Space** | TransE | 0.310 | 0.218 | 0.345 | 0.495 | 0.232 | 0.061 | 0.366 | 0.522 |
| | DistMult | 0.342 | 0.249 | 0.378 | 0.531 | 0.451 | 0.414 | 0.466 | 0.523 |
| | RotatE | 0.338 | 0.241 | 0.375 | 0.533 | 0.476 | 0.428 | 0.492 | 0.571 |
| | TuckER | 0.358 | 0.266 | 0.394 | 0.544 | 0.470 | 0.443 | 0.482 | 0.526 |
| | RulE | 0.362 | 0.266 | 0.400 | 0.553 | 0.519 | 0.475 | 0.538 | 0.605 |
| | MetaSD | 0.391 | 0.300 | 0.428 | 0.571 | 0.491 | 0.447 | 0.504 | 0.570 |
| **Large Language Model** | KGT5 | 0.276 | 0.210 | - | 0.414 | 0.508 | 0.487 | - | 0.544 |
| | SimKGC | 0.336 | 0.249 | 0.362 | 0.511 | **0.666** | **0.587** | **0.717** | **0.800** |
| | N-Former | 0.373 | 0.279 | 0.412 | 0.556 | 0.489 | 0.446 | 0.504 | 0.581 |
| **Graph Neural Network** | CompGCN | 0.355 | 0.264 | 0.39 | 0.535 | 0.479 | 0.443 | 0.494 | 0.546 |
| | RED-GNN | 0.374 | 0.282 | - | 0.589 | 0.519 | 0.465 | - | 0.602 |
| | ULTRA | 0.368 | 0.272 | - | 0.564 | 0.480 | 0.414 | - | 0.614 |
| | NBFNet | 0.415 | 0.321 | 0.454 | 0.599 | 0.551 | 0.497 | 0.573 | 0.666 |
| **Relational Attention** | HittER | 0.373 | 0.279 | 0.409 | 0.558 | 0.503 | 0.462 | 0.516 | 0.584 |
| | MGTCA | 0.393 | 0.291 | 0.428 | 0.583 | 0.511 | 0.475 | 0.525 | 0.593 |
| | KnowFormer | 0.430 | **0.343** | - | 0.608 | 0.579 | 0.528 | - | 0.687 |
| **Hyperbolic Space** | MuRP | 0.335 | 0.243 | 0.367 | 0.518 | 0.481 | 0.440 | 0.495 | 0.566 |
| | ATTH | 0.351 | 0.255 | 0.386 | 0.543 | 0.490 | 0.443 | 0.508 | 0.581 |
| | FHRE | 0.345 | 0.255 | 0.375 | 0.528 | 0.494 | 0.458 | 0.510 | 0.563 |
| **Ours** | HyPNet | **0.431** | 0.340 | **0.467** | **0.611** | 0.565 | 0.510 | 0.594 | 0.699 |

HyPNet outperformed on all eight metrics compared to all Euclidean space-based methods, Hyperbolic space-based methods, large language model-based methods, and graph neural network-based methods, apart from the WN18RR performance of SimKGC. WN18RR is a knowledge base extracted from WordNet, which contains many common words and lexical relations between them. The entities and relations in WordNet have clear semantics and are widely present in the training texts of large language models. Rich descriptions enable LLM to achieve the best performance on WN18RR.

Notably, compared with the base model NBFNet, HyPNet improved the MRR metric by 1.6 and 0.9 on the two types of datasets, respectively. Compared to another comparable relational attention-based method, Knowformer, we surpassed it on FB15k-237 using nearly half the number of parameters (approximately 300M vs 600M). However, on WN18RR, we only exceeded it in terms of Hit@10. This may be due to the smaller size of the WN18RR dataset, as the overall hierarchical relationships are not as complex as those in FB15k-237.

## 5.4 INDUCTIVE PERFORMANCE

We follow the 8 versions of standard inductive datasets (Teru et al., 2020) from FB15k-237 and WN18RR. As shown in Table 2, HyPNet achieved competitive results on the FB15k-237 dataset, but the performance on WN18RR was not satisfactory. HyPNet achieved the best results in all Hit@10 metrics for all versions of FB15k-237 and the v1 version of WN18RR.

We were unable to achieve baseline results comparable to those reported in the original paper with † under the current software environment. Specifically, during the training process of the inductive setting WN18RR, we observed multiple instances where the validation MRR is high but the test MRR is low. This may be due to the risk of over-fitting associated with the complex hyperbolic space when the dataset is small. Techniques such as weight decay or reducing curvature do not fully address this phenomenon.

Table 2: Inductive Multi-relational Link Prediction. Best results are bold, second results are underlined. In the inductive setting of NBFNet on WN18RR, † indicates the baseline results that we replicated.

| Method | v1 | | | v2 | | | v3 | | | v4 | | |
|---|---|---|---|---|---|---|---|---|---|---|---|---|
| | MRR | H@1 | H@10 | MRR | H@1 | H@10 | MRR | H@1 | H@10 | MRR | H@1 | H@10 |
| **FB15k-237** | | | | | | | | | | | | |
| DRUM | 0.333 | 0.247 | 0.474 | 0.395 | 0.284 | 0.595 | 0.402 | 0.308 | 0.571 | 0.410 | 0.309 | 0.593 |
| RED-GNN | 0.369 | 0.302 | 0.483 | 0.469 | 0.381 | 0.629 | 0.445 | 0.351 | 0.503 | 0.442 | 0.340 | 0.621 |
| A*Net | 0.457 | **0.381** | 0.589 | 0.510 | 0.419 | 0.672 | 0.476 | 0.389 | 0.629 | 0.466 | 0.365 | 0.645 |
| AdaProp | 0.310 | 0.191 | 0.551 | 0.471 | 0.372 | 0.659 | 0.471 | 0.377 | 0.637 | 0.454 | 0.353 | 0.638 |
| Ingram | 0.293 | 0.167 | 0.493 | 0.274 | 0.163 | 0.482 | 0.233 | 0.140 | 0.408 | 0.214 | 0.114 | 0.397 |
| NBFNet | 0.442 | 0.335 | 0.574 | 0.514 | 0.421 | 0.685 | 0.476 | 0.384 | 0.637 | 0.453 | 0.360 | 0.627 |
| KnowFormer | **0.466** | 0.378 | 0.606 | **0.532** | **0.433** | 0.703 | **0.494** | **0.400** | 0.659 | **0.480** | **0.383** | 0.653 |
| HyPNet | 0.463 | 0.372 | **0.619** | 0.517 | 0.408 | **0.705** | 0.485 | 0.391 | **0.662** | 0.471 | 0.356 | **0.669** |
| **WN18RR** | | | | | | | | | | | | |
| DRUM | 0.666 | 0.613 | 0.777 | 0.646 | 0.595 | 0.747 | 0.380 | 0.330 | 0.477 | 0.627 | 0.586 | 0.702 |
| RED-GNN | 0.701 | 0.653 | 0.799 | 0.690 | 0.633 | 0.780 | 0.427 | 0.368 | 0.524 | 0.651 | 0.606 | 0.721 |
| AdaProp | 0.733 | 0.668 | 0.806 | 0.715 | 0.642 | 0.826 | 0.474 | 0.396 | 0.588 | 0.662 | 0.611 | 0.755 |
| Ingram | 0.277 | 0.130 | 0.606 | 0.236 | 0.112 | 0.480 | 0.230 | 0.116 | 0.466 | 0.118 | 0.041 | 0.259 |
| SimKGC | 0.315 | 0.192 | 0.567 | 0.378 | 0.239 | 0.650 | 0.303 | 0.186 | 0.543 | 0.308 | 0.175 | 0.577 |
| A*Net | 0.727 | 0.682 | 0.810 | 0.704 | 0.649 | 0.803 | 0.441 | 0.386 | 0.544 | 0.661 | 0.616 | 0.743 |
| KnowFormer | 0.739 | 0.702 | 0.806 | 0.697 | 0.651 | 0.776 | 0.467 | 0.406 | 0.571 | 0.646 | 0.609 | 0.727 |
| NBFNet † | 0.701 | 0.622 | 0.827 | 0.648 | 0.569 | 0.778 | 0.429 | 0.355 | 0.548 | 0.590 | 0.527 | 0.690 |
| HyPNet | 0.716 | 0.629 | 0.841 | 0.668 | 0.589 | 0.786 | 0.398 | 0.330 | 0.511 | 0.623 | 0.546 | 0.739 |

## 5.5 ABLATION STUDY

**Number of Layers** $n$ Figure 1 conducts a statistical analysis about the shortest paths to the tail entities in the test set of FB15k-237 and found that almost all paths are within 8 hops. The results of the Table 3a show that HyPNet prevents the over-smoothing phenomenon, and the increase in MRR is positively correlated with the statistics of the shortest paths in the test set. In different experiments, $\gamma$ can achieve the best results within the range of 0.1 to 0.5.

**Value of** $\gamma$ The scale factor $\gamma > 0$ measures the matrix properties of $A$. As $\gamma > 0$ approaches 0, $A$ becomes closer to an antisymmetric matrix; a larger $\gamma$ indicates that the decay rate of the feature norm is faster. In different experiments, $\gamma$ can achieve the best results within the range of 0.1 to 0.5.

**Relation Category Performance** There are four relation category, *one-to-one*, *one-to-many*, *many-to-one*, and *many-to-many*, and two query category head query $(?, r, t)$ and tail query $(h, r, ?)$. [1] Regarding the MRR perfor-

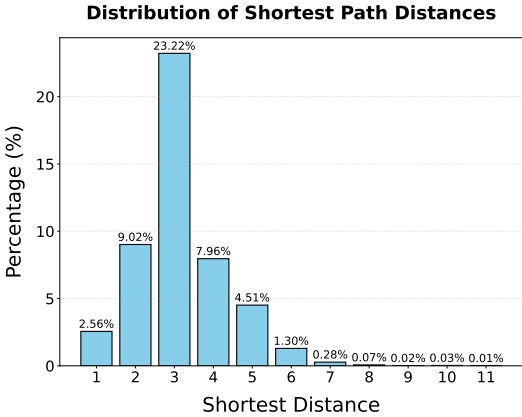

Figure 1: Statistics of the shortest distance from query to the answer tail entity in the FB15k-237 test set. In the WN18RR dataset, almost all shortest path distances are within 2.

---

[1]Relation category focus on the average number of tails per head and the average number of heads per tail. The category is *one* if the average number is smaller than 1.5 and *many* otherwise.

mance of FB15k-37, we can observe in Table 3c that, except for a slight decline in the 1-1 Relation, all other metrics have significantly increased, especially A and B. Both of these relation types represent multiple outgoing edges that share the relationship, indicating that hyperbolic space path aggregation alleviates the effects of overcrowding. However, the results for 1-1 suggest that the current framework is not fully compatible with Euclidean space, and exploring how to integrate spaces is a meaningful avenue for further research.

For the learnable curvature, the curvature of each relation will ultimately oscillate around a certain value. The parameters have little impact on the model's performance, possibly because it is sufficient to ensure the relative magnitude of the scores. The numerical value of the curvature only needs to meet the computational precision requirements to prevent issues in training caused by the curvature being too small or negative.

Table 3: Ablation study of FB15k-237 MRR performance.

| Method | Layer number ($n$) | | | |
|---|---|---|---|---|
| | 2 | 4 | 6 | 8 |
| HyPNet | 0.336 | 0.384 | 0.431 | 0.432 |

(a) Different number of layers.

| Method | Scale factor ($\gamma$) | | | |
|---|---|---|---|---|
| | 0.1 | 0.3 | 0.5 | 0.7 |
| HyPNet | 0.430 | 0.428 | 0.431 | 0.420 |

(b) Different values of balance.

| Method | Relation Category | | | |
|---|---|---|---|---|
| | 1-to-1 | 1-to-N | N-to-1 | N-to-N |
| TransE | 0.498/0.488 | 0.455/0.071 | 0.079/0.744 | 0.224/0.330 |
| RotatE | 0.487/0.484 | 0.467/0.070 | 0.081/0.747 | 0.234/0.338 |
| NBFNet | 0.578/0.600 | 0.499/0.122 | 0.165/0.790 | 0.348/0.456 |
| HyPNet | 0.570/0.588 | **0.506/0.144** | **0.206/0.794** | **0.358/0.470** |

(c) MRR Performance by relation category. The slash distinguishes two queries (?,r,t) and (h,r,?) formed from the same triple.

# 6 CONCLUSION

We propose HyPNet that combines the advantages of hyperbolic space for hierarchical relations and path aggregation to prevent over-smoothing. We extended the antisymmetric matrix for stable multi-layer GNN training into hyperbolic space. HyPNet extracts the widely existing *one-to-many*, *many-to-one*, and *many-to-many* relationships (tree structures) in the graph. HyPNet achieved results comparable to the previous state-of-the-art on the FB15k-237 and WN18RR standard datasets in both transductive and inductive settings, while using only half the parameters.

# 7 LIMITATION

HyPNet has two notable limitations. First, the convergence rates of the model parameters and trainable curvatures are inconsistent, which results in high variance in the learning outcomes on some smaller inductive setting datasets, making it difficult to surpass previous work. Second, although the model utilizes hyperbolic space to model hierarchical relationships, it cannot adapt to other types of curvature. In the future, we hope to expand the range of curvature, integrate different types of relationships, and innovate the rewiring methods of graph structures.

## 8 ETHICS STATEMENT

We ensure that our submission will not raise questions regarding the Code of Ethics.

## 9 REPRODUCIBILITY STATEMENT

The complete source code for HyPNet is publicly available at the anonymous repository: `https://anonymous.4open.science/r/anonymous_HyPNet-B0B0`. We implement HyPNet in Pytorch 2.8.0, CUDA 12.8, Python 3.12.3, using 4 NVIDIA GeForce RTX 5090 GPUs with 32GB of Memory and an INTEL(R) XEON(R) GOLD 6530 CPU @2.6GB. Main dependencies explicitly listed in a *README.md* file within the repository.

By providing explicit model formulation, public code with clear dependencies, and standardized datasets, we ensure that HyPNet's results can be fully replicated by the research community.

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

## A USE OF LLMS

LLM is used solely for translation, and the paper does not utilize any directly generated text.

## B HYPERBOLIC GEOMETRY DETAILS

The Poincaré ball model is one of five *isometric* models of hyperbolic geometry (Cannon et al.,
1997). The *Riemannian metric* of poincaré ball $g^{\mathbb{B}}$ is conformal to Euclidean metric $g^{\mathbb{R}} = \boldsymbol{I}$, where
$g^{\mathbb{B}} = (\lambda_{\boldsymbol{x}}^c)^2 g^{\mathbb{R}}$ with a factor $\lambda_{\boldsymbol{x}}^c = 2/(1 - c\|\boldsymbol{x}\|^2)$.

Defined by the HNN (Ganea et al., 2018), the exponential map and logarithmic map at any point on
the poincaré ball $\boldsymbol{x} \in \mathbb{B}_c^d$ and the corresponding tangent space $\mathcal{T}_{\boldsymbol{x}}\mathbb{B}_c^d$,

$$\exp_{\mathbf{x}}^c(\mathbf{v}) = \mathbf{x} \oplus_c \left( \tanh\left( \sqrt{c}\frac{\lambda_{\mathbf{x}}^c \|\mathbf{v}\|}{2} \right) \frac{\mathbf{v}}{\sqrt{c}\|\mathbf{v}\|} \right), \tag{13}$$

$$\log_{\mathbf{x}}^c(\mathbf{y}) = \frac{2}{\sqrt{c}\lambda_{\mathbf{x}}^c} \tanh^{-1}(\sqrt{c}\| - \mathbf{x} \oplus_c \mathbf{y}\|) \frac{-\mathbf{x} \oplus_c \mathbf{y}}{\| - \mathbf{x} \oplus_c \mathbf{y}\|}. \tag{14}$$

The generalization of a *bias translation* in the poincaré ball is naturally given by moving along
geodesics. From the perspective of parallel transport, Mobius translation of a point $\boldsymbol{x} \in \mathbb{B}_c^d$ by a
bias $\boldsymbol{b} \in \mathcal{T}_{\boldsymbol{0}}\mathbb{B}_c^d$ is given by

$$\boldsymbol{x} \oplus_c \boldsymbol{b} = \exp_{\boldsymbol{x}}^c(P_{\boldsymbol{0}}^c \to \boldsymbol{x}(\log_{\boldsymbol{0}}^c(b))) = \exp_{\boldsymbol{x}}^c \left( \frac{\lambda_{\boldsymbol{0}}^c}{\lambda_{\boldsymbol{x}}^c} \log_{\boldsymbol{0}}^c(\boldsymbol{b}) \right). \tag{15}$$

## C PROOF OF THEOREM 1

**Lemma 1** (I,ax equivalence theorem (Lax & Richtmyer, 1956)). *A well-posed initial value problem
of a system of linear partial differential equations, a consistent linear differential equation converges
if and only if the equation is stable.*

The difference equation $\frac{d\boldsymbol{z}}{dt} = A\boldsymbol{z} + M\boldsymbol{x} + b$ compatible with the differential equation $\boldsymbol{z}^n =
(I + \Delta t \cdot \boldsymbol{A})\boldsymbol{z}^{n-1} + \Delta t \cdot \boldsymbol{B}\boldsymbol{x}^{n-1} + \Delta t \boldsymbol{c}$. When the $\Delta t = 1$, it is the update function defined in
Euclidean space Eq. (10).

**Definition 1** (Stablity of Ordinary Difference Equation). *Giver a linear ODE $\boldsymbol{z}'(t) = \boldsymbol{A}\boldsymbol{z}(t)$, a
solution $\boldsymbol{z}(t)$ of the ODE with initial condition $\boldsymbol{z}(0)$ is stable if for any $\epsilon > 0$, there exists a $\delta > 0$
such that any other solution $\tilde{\boldsymbol{h}}(t)$ of the ODE with initial condition $\tilde{\boldsymbol{h}}(0)$ satisfying $|\boldsymbol{z}(0) - \tilde{\boldsymbol{z}}(0)| \leq \delta$
also satisfies $|\boldsymbol{z}(t) - \tilde{\boldsymbol{z}}(t)| \leq \epsilon$, for all $t \geq 0$.*

If $\boldsymbol{A}$ is diagonalizable, i.e. $\boldsymbol{P}^{-1}\boldsymbol{A}\boldsymbol{P} = \boldsymbol{\Lambda}$, where $\boldsymbol{\Lambda}$ is a diagonal matrix of the eigenvalues of $\boldsymbol{A}$
and the columns of $\boldsymbol{P}$ are the corresponding eigenvectors.

$$|\boldsymbol{z}(t) - \tilde{\boldsymbol{z}}(t)| = |e^{\boldsymbol{A}t}(\boldsymbol{z}(0) - \tilde{\boldsymbol{z}}(0))| = e^{\boldsymbol{P}Re(\boldsymbol{\Lambda})\boldsymbol{P}^{-1}t}|\boldsymbol{z}(0) - \tilde{\boldsymbol{z}}(0)|. \tag{16}$$

If $\max_{1,2,\ldots,d} Re(\boldsymbol{\Lambda}) <= 0$, the ODE is stable.

**Lemma 2** (Stablity of Neural Deep Graph Network (Gravina et al., 2023)). *The solution of an ODE
$\boldsymbol{z}'(t) = f_{\mathcal{G}}(\boldsymbol{z}(t)) = \sigma(\boldsymbol{A}z(t) + \Phi(\boldsymbol{Z}(t)) + \boldsymbol{b}_t)$ is stable and non-dissipative if*

$$\forall i = 1, 2, \ldots, n, \quad Re(\lambda_i(\boldsymbol{J}(t))) = 0, \tag{17}$$

*where $\boldsymbol{J}(t) \in \mathbb{R}^{n \times n}$ be the Jacobian matrix of $f$, and $\lambda_i$ denotes the $i$-th eigenvalue. $Re(\cdot)$ denotes
the real part of a complex number. $\boldsymbol{J}(t)$ does not change significantly over time.*

If weight is anti-symmetric $\boldsymbol{A} = \boldsymbol{W} - \boldsymbol{W}^T$, $\boldsymbol{J}(t)$ can be divided into two parts as follows,

$$\boldsymbol{J}(t) = \mathcal{D}_{\text{test}} diag\left[\sigma'\left((\boldsymbol{W} - \boldsymbol{W}^T)\boldsymbol{z}(t) + \boldsymbol{V}\boldsymbol{z}(t) + \boldsymbol{b}\right)\right](\boldsymbol{W} - \boldsymbol{W}^T), \tag{18}$$

First, the activation function derivative part guarantees to be within $[0, 1]$ regardless of whether tanh or ReLU, thus $\boldsymbol{J}(t)$ does not change significantly over time. The second anti-symmetric matrix satisfies the $Re(\lambda_i(\boldsymbol{A}(t))) = 0$.

**Lemma 3** (Hyperbolic Triangle Inequality). *In poincaré ball model, the logarithmic mapping has an identity relationship with hyperbolic distance.*

$$d_c^{\mathbb{B}}(-\boldsymbol{u}, \boldsymbol{w}) = d_c^{\mathbb{B}}(\boldsymbol{0}, -\boldsymbol{u} \oplus_c \boldsymbol{w}) = 2\|log_{\boldsymbol{0}}^c(-\boldsymbol{u} \oplus_c \boldsymbol{w}))\|. \tag{19}$$

*Geodesics are the shortest distances in hyperbolic space and still satisfy the triangle inequality. Given $\boldsymbol{u}, \boldsymbol{v}, \boldsymbol{w} \in \mathbb{B}_c^d$,*

$$d_c^{\mathbb{B}}(-\boldsymbol{u}, \boldsymbol{w}) \leq d_c^{\mathbb{B}}(-\boldsymbol{u}, \boldsymbol{v}) + d_c^{\mathbb{B}}(-\boldsymbol{v}, \boldsymbol{w}), \tag{20}$$

*Given $\boldsymbol{x}, \boldsymbol{y} \in \mathcal{T}_{\boldsymbol{0}}\mathbb{B}_c^d$, another version is as follws,*

$$\|\log_{\boldsymbol{0}}^c(\exp_{\boldsymbol{0}}^c(\boldsymbol{x}) \oplus_c \exp_{\boldsymbol{0}}^c(\boldsymbol{y}))\| \leq \|\log_{\boldsymbol{0}}^c(\exp_{\boldsymbol{0}}^c(\boldsymbol{x}))\| + \|\log_{\boldsymbol{0}}^c(\exp_{\boldsymbol{0}}^c(\boldsymbol{y}))\| = \|\boldsymbol{x}\| + \|\boldsymbol{y}\|. \tag{21}$$

*The above inequality degenerates into an equality when points $\boldsymbol{x}$ and $\boldsymbol{y}$ are collinear.*

**Theorem 1.** *For the update function defined in Euclidean space*

$$\boldsymbol{z}^n = \boldsymbol{z}^{n-1} + \sigma(A\boldsymbol{z}^{n-1} + \Phi(\boldsymbol{M}(x)) + \boldsymbol{b}), x \in \mathcal{E}(h), \quad t \notin \mathcal{E}(h). \tag{22}$$

*If $\boldsymbol{A}$ is an anti-symmetric matrix, the solution $\tilde{\boldsymbol{z}}^n$ of the update function Eq. (10) converges. Furthermore, the update function defined in the Hyperbolic space Eq. (9) also satisfies. The implementation using Euclidean operators serves as an upper bound for the implementation using hyperbolic operators.*

**Proof 1.** *Lemma 1 shows that our update function defined in Eudiean Eq. (10) is compatible with the corresponding ordinary differential equation with $\Delta t = 1$, and Lemma 2 demonstrates that the corresponding ordinary differential equations are stable and non-dissipative. The combination of the first two lemmas indicates that Eq. (10) is convergent.*

$$\log_{\boldsymbol{0}}^{c_q}\left(\exp_{\boldsymbol{0}}^{c_q}(\boldsymbol{A}\boldsymbol{z}_q^{n-1}) \oplus \exp_{\boldsymbol{0}}^{c_q}(\Phi(\boldsymbol{M}_q^n(x))) \oplus \exp_{\boldsymbol{0}}^{c_q}(\boldsymbol{b})\right) \leq \boldsymbol{A}\boldsymbol{z}_q^{n-1} + \Phi(\boldsymbol{M}_q(x)) + \boldsymbol{b} \tag{23}$$

*Lemma 3 indicates that the norm of the update function implemented in hyperbolic space Eq. (9) has the upper bound as the norm of Eq. (10), essentially stating that hyperbolic space and Euclidean space of the same dimension are homeomorphic manifolds.*

# D  DATASETS STATIC

**WN1818RR** (Dettmers et al., 2018) is a static KG extracted from WordNet, which contains the common words and lexical relations between them.

**FB15k-237** (Toutanova et al., 2015) is a static KG extracted from Freebase, a large Knowledge base that uses web links to connect real-world events.

The inductive setting means train on one graph and test on another graph. We follow the 8 version extracted from FB15k-237 and WN18RR (Teru et al., 2020).

Table 4: Dataset Statistics for transductive knowledge graph reasoning datasets.

| Dataset | #Relation | #Entity | #Triplet | | |
|---|---|---|---|---|---|
| | | | #Train | #Valid | #Test |
| FB15k-237 | 237 | 14,541 | 272,115 | 17,535 | 20,466 |
| WN18RR | 11 | 40,943 | 86,835 | 3,034 | 3,134 |

Table 5: Dataset Statistics for inductive knowledge graph reasoning datasets. In each split, one needs to infer #Query triplets based on #Fact triplets.

| Dataset | | #Relation | Train | | | Validation | | | Test | | |
|---|---|---|---|---|---|---|---|---|---|---|---|
| | | | #Entity | #Query | #Fact | #Entity | #Query | #Fact | #Entity | #Query | #Fact |
| FB15k-237 | v1 | 180 | 1,594 | 4,245 | 4,245 | 1,594 | 489 | 4,245 | 1,093 | 205 | 1,993 |
| | v2 | 200 | 2,608 | 9,739 | 9,739 | 2,608 | 1,166 | 9,739 | 1,660 | 478 | 4,145 |
| | v3 | 215 | 3,668 | 17,986 | 17,986 | 3,668 | 2,194 | 17,986 | 2,501 | 865 | 7,406 |
| | v4 | 219 | 4,707 | 27,203 | 27,203 | 4,707 | 3,352 | 27,203 | 3,051 | 1,424 | 11,714 |
| WN18RR | v1 | 9 | 2,746 | 5,410 | 5,410 | 2,746 | 630 | 5,410 | 922 | 188 | 1,618 |
| | v2 | 10 | 6,954 | 15,262 | 15,262 | 6,954 | 1,838 | 15,262 | 2,757 | 441 | 4,011 |
| | v3 | 11 | 12,078 | 25,901 | 25,901 | 12,078 | 3,097 | 25,901 | 5,084 | 605 | 6,327 |
| | v4 | 9 | 3,861 | 7,940 | 7,940 | 3,861 | 934 | 7,940 | 7,084 | 1,429 | 12,334 |

# E  OVER-SQUASHING IN KG

*Balanced Forman curvature* is a metric for measuring the over-squashing in the graph. In line with the discussion about geodesic dispersion, one expects $\sharp_\triangle$ to be related to positive curvature (complete graph), $\sharp_\square^i$ to zero curvature (grid), and the remaining *outgoing* edges to negative curvature (tree). The Balanced Forman curvature is related to the Jacobian of the hidden features.

**Definition 2** (**Balanced Forman curvature** (Topping et al., 2022))**.** *For any edge $i \sim j$ in a simple, unweighted graph $\mathcal{G}$, we let $Ric(i,j)$ be zero if $\min\{d_i, d_j\} = 1$ and otherwise*

$$Ric(i,j) := \frac{2}{d_i} + \frac{2}{d_j} - 2 + 2\frac{|\sharp_\triangle(i,j)|}{\max\{d_i,d_j\}} + \frac{|\sharp_\triangle(i,j)|}{\min\{d_i,d_j\}} + \frac{(\lambda_{\max})^{-1}}{\max\{d_i,d_j\}}\left(|\sharp_\square^i| + |\sharp_\square^j|\right),$$

(24)

*where the last term is set to be zero if $|\sharp_\square^i|$ (and hence $|\sharp_\square^j|$) is zero. In particular $Ric(i,j) > -2$.*

In the case of heterogeneous graphs, such as knowledge graphs, the presence of different types of relations and entities of the same relation both increases the possibility of over-squashing. Figure 2

| Metric | Dataset(Layer Comparison) | | | |
|---|---|---|---|---|
| | Cora | Pubmed | Disease | Airport |
| ROP | 0.94/0.86 | 0.95/0.86 | 0.61/0.48 | 0.90/0.88 |
| AP | 0.94/0.88 | 0.95/0.87 | 0.58/0.49 | 0.90/0.90 |

Table 6: Comparison of different layers of HGCN on Link prediction results for homogeneous graphs.

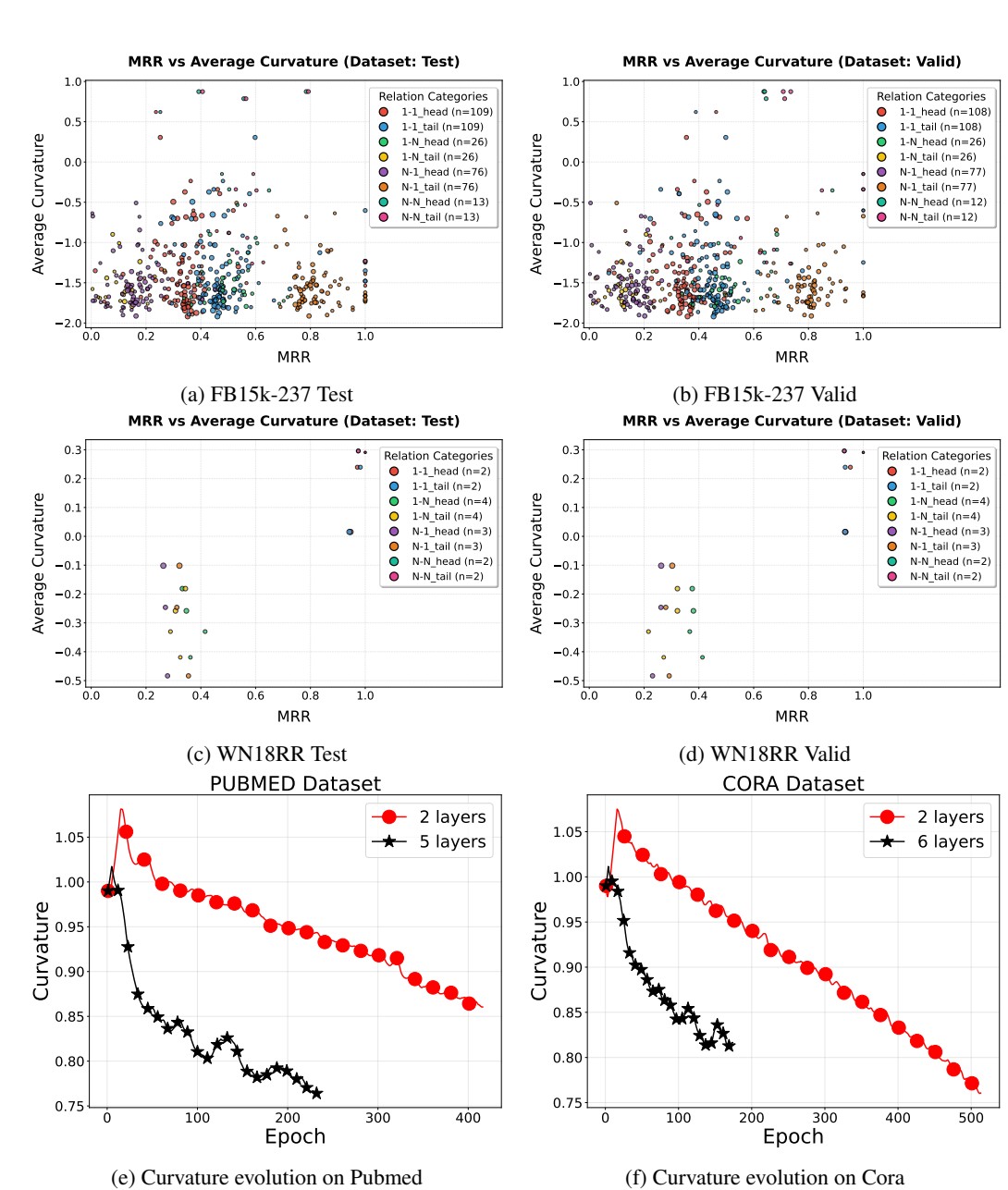

(a) FB15k-237 Test

(b) FB15k-237 Valid

(c) WN18RR Test

(d) WN18RR Valid

(e) Curvature evolution on Pubmed

(f) Curvature evolution on Cora

Figure 2: The first four figures illustrate the relationship between MRR performance and curvature across different relation categories, with results derived from NBFNet, indicating the presence of a substantial number of tree structures in the knowledge graph (KG). The last two figures show the rate of curvature decrease during the training phase of HGCN, revealing a converging trend before early stopping. All subfigures indicate that both path aggregation methods and hyperbolic space approaches face challenges related to oversmoothing.

