# OpenReview forum: "Hyperbolic Space Path Aggregation for Over-squashing in Multi-relational Graph"
_ICLR.cc/2026/Conference — ICLR 2026 Conference Withdrawn Submission_

### Official Review · Reviewer_pPQC · 2025-10-24

**Soundness:** 2
**Presentation:** 2
**Contribution:** 2
**Rating:** 2
**Confidence:** 4

**Summary:**

This paper proposes HyPNet, a path-aggregation model for multi-relational link prediction that runs the NBFNet-style reasoning pipeline in hyperbolic space using Möbius operators. The encoder employs an anti-symmetric update to stabilize deeper stacks and allegedly mitigate over-squashing. The decoder adds a hyperbolic distance term in the Poincaré ball on top of a 2-layer MLP. Reported results cover FB15k-237 and WN18RR (standard transductive) plus eight inductive splits, with HyPNet outperforming path-based and hyperbolic baselines on FB15k-237 and being competitive on WN18RR and their inductive variants.

**Strengths:**

- Conceptually clean fusion of path aggregation and hyperbolic geometry with a consistent operator set and clear presentation of the needed hyperbolic preliminaries for those unfamiliar with hyperbolic geometry.
- The anti-symmetric update is well-motivated by stability arguments (Theorem 1 + Appendix C), aligning with known deep-GNN stability intuitions.
- Some analysis toward relation categories (1-to-N / N-to-1) and a pointer to Balanced Forman curvature as a good point out: not much literature has studied about oversquashing effect in knowledge graphs.

- Code is provided in an anonymous repo with dependency details and datasets

**Weaknesses:**

Major:

- **Unclear novelty & support for core claims.** The paper’s main step is to modify NBFNet into hyperbolic space with an anti-symmetric update. The paper claims (i) alleviating over-squashing (line 72), (ii) enhancing expressive power(line 75), (iii) stabilizing curvature, and that prior methods suffer “faster curvature collapse” (line 65) and (iv) mitigating over-smoothing (line 67), all of which are central to the motivation. But these points are asserted with limited direct empirical or theoretical evidence in the main text. Oversquashing is not measured with standard proxies (e.g. sensitivity analysis [1]), and curvature-collapse evidence is deferred to Appendix E. The “enhanced expressive power” claim lacks a theorem to show the separation power or a targeted controlled experiment with possibly synthetic datasets.

- **Limited datasets & scope.** Results are reported only on FB15k-237 and WN18RR (with inductive variants). Given the paper’s general claims about hierarchy and squashing, a broader selection of datasets (e.g., datasets with different degree/diameter/relations) or a synthetic suite designed to stress oversquashing, expressivity, and hierarchy would be more convincing. You can select them from ULTRA paper [2]


- **Ablations and incorrect definition of hierarchical**. The ablation studies feel like hyperparameter tuning rather than testimony ofthe  paper's major claim for the advantages of the proposed method. There is no **Euclidean vs. hyperbolic** controlled comparison at matched capacity to isolate the geometry from other factors. Also, the paper argues that hyperbolic helps “hierarchical structure,” but using improvements in 1-to-N / N-to-1 categories as evidence. The author should either construct synthetic datasets using relation sets that satisfy r(x,y) => t(x,y) as a proper "hierarchy" definition [3] of relations to show, or not make the claim of "hierarchical" and stick with 1-to-N/N-to-1 arguments.

- **Limited empirical evidence**.  The overall performance of HyPNet does not seem too beneficial. Although not reaching SoTA on all datasets is acceptable, given that the author does not provide a study on "why" it is the case but just possible speculation, it raises concerns about the general applicability of this method.

- **Presentation issues**.
  - It is hard to visualize how the Euclidean (NBFNet) vs the hyperbolic HyPNet. Can the author include a figure to describe how this works to aid the understanding? The author can ignore this comment if the rebuttal does not allow uploading additional figures.
  - Table 1 mixes LLM-based methods (that consume text) with pure-graph methods. The paper partly acknowledges why SimKGC excels on WN18RR, but the training signal parity is not spelled out; please state clearly what inputs each method receives.
  - Additionally, in the GNN-based method, ULTRA is a foundation model trained with a different pretraining mix, so the input to the model is potentially different from other baselines, creating an apples-and-oranges comparison. Please clarify what pretraining data mix is the baseline chosen for a fair comparison.



Minor:

- “HyppPath → HyPNet” (and generally HypPath vs HyPNet inconsistency).

- “Line 71: E”.

- A few places write FB15k-37 rather than FB15k-237 (e.g., Table 3 caption/text). Please unify.

- In Table 2, FB15k-237 best/second-best are highlighted, but WN18RR highlights appear inconsistent/absent.

- Standard deviation of the experiments are not reported

[1] Topping J, Di Giovanni F, Chamberlain B P, et al. Understanding over-squashing and bottlenecks on graphs via curvature[J]. arXiv preprint arXiv:2111.14522, 2021.

[2] Galkin M, Yuan X, Mostafa H, et al. Towards foundation models for knowledge graph reasoning[J]. arXiv preprint arXiv:2310.04562, 2023.

[3] Abboud R, Ceylan I, Lukasiewicz T, et al. Boxe: A box embedding model for knowledge base completion[J]. Advances in Neural Information Processing Systems, 2020, 33: 9649-9661.

**Questions:**

- Can the author move the critical evidence out of the appendix and provide quantitative diagnostics for quantifying over-squashing, which is the main claim of the paper?

- The author claims prior hyperbolic GNNs collapse curvature faster with depth; can you bring those Appendix E plots into the main paper?

- The paper itself notes multiple runs with high validation MRR but low test MRR and speculates small-data hyperbolic overfitting; please add regularization/bias-control ablations to confirm the reason.

 - Why is the additional hyperbolic distance term needed beyond the MLP? What role does it play relative to the MLP output (regularization, geometric bias)?

- Can you also report the standard deviation?

---

### Official Review · Reviewer_iSsm · 2025-10-27

**Soundness:** 3
**Presentation:** 3
**Contribution:** 2
**Rating:** 4
**Confidence:** 3

**Summary:**

The paper proposes HyPNet, a framework that integrates path-based message passing with hyperbolic geometry for multi-relational link prediction. It aims to alleviate over-squashing and curvature collapse, two critical issues in deep hyperbolic GNNs. HyPNet performs relational path aggregation directly in the Poincaré ball using Möbius operators, while employing an anti-symmetric update matrix to ensure training stability across layers. The model combines hyperbolic distance with a Euclidean decoder for scoring triplets. Experiments on both transductive (FB15k-237, WN18RR) and inductive datasets show improvements over path-based (e.g., NBFNet) and hyperbolic methods (e.g., MuRP, ATTH), while using fewer parameters. Ablation studies suggest that the anti-symmetric update and hyperbolic path formulation help maintain curvature stability and improve performance on complex relation categories.

**Strengths:**

- Original combination of path aggregation and hyperbolic representation: the idea of performing path reasoning directly in the hyperbolic manifold is novel and well-motivated by the complementary strengths of the two paradigms.
- Sound mathematical formulation: the paper rigorously defines Möbius operations and provides a stability proof (Theorem 1) showing convergence of the anti-symmetric update in both Euclidean and hyperbolic settings.
- Addresses real training issues,  notably curvature collapse and over-squashing, with a theoretically grounded mechanism rather than ad-hoc normalization.
- Comprehensive evaluation: results cover both transductive and inductive settings with a broad range of baselines (Euclidean, hyperbolic, attention-based, LLM-based).
- Ablation studies (varying layers, γ, and relation categories) give useful insight into when hyperbolic path aggregation provides gains.

**Weaknesses:**

- Clarity and section flow: In some sections, the logical flow is hard to follow due to the lack of introductory context or transition sentences.
Briefly outlining the purpose of each section would make the paper easier to read and help situate the reader within the technical progression.
- Conceptual clarity: while the paper mixes path aggregation and hyperbolic embedding, it is not fully clear why performing path aggregation in hyperbolic space alleviates over-squashing beyond empirical observation. A more explicit geometric intuition  would strengthen the argument .
- Limited novelty of components: many elements (Möbius operations, antisymmetric updates) are adapted from prior works (e.g., HGCN, AntiSymDGN). The true novelty lies in combining them, but this could be made clearer and contrasted to previous mixed-geometry works such as MGTCA or FHRE.
- Empirical gains are modest on some datasets (especially WN18RR) and inconsistent in the inductive setting; performance drops suggest hyperbolic modeling may still overfit on small graphs.
- Fair comparison issue: Several baseline results in Table 1 appear to be taken from prior papers (as indicated by “–”), suggesting that not all methods were re-run under the same settings.
- Missing variance reporting: Results are reported as single numbers without mean ± std over multiple seeds. Given the sensitivity of KGC methods to random initialization, negative sampling, and early stopping, reporting averages and standard deviations across ≥3–5 seeds is essential for statistical validity and fair comparison
For a fair comparison, key baselines should ideally be re-executed with identical seeds, splits, and evaluation protocols; otherwise, differences may partly stem from random or implementation factors rather than true model gains
- Complexity analysis missing: although the method scales linearly with path length, there is no clear runtime or memory comparison against NBFNet or hyperbolic GNNs.
- Missing link to broader GNN literature:The paper focuses mainly on link-prediction models, overlooking multi-relational GNNs developed for node classification (e.g., [1,4]) that could also provide useful node embeddings for KGC.
Referencing or briefly testing such models would better situate this work within the broader multi-relational GNN landscape.

[1] Schlichtkrull et al., Modeling relational data with graph convolutional networks, 2018

[2]  Ferrini et al., Meta-Path Learning for Multi-relational Graph Neural Networks, LOG 2023

[3] Zhu et al., Relation structure-aware heterogeneous graph neural network. 2019

[4] Ferrini et al., A Self-Explainable Heterogeneous GNN for Relational Deep Learning, TMLR 2025

**Questions:**

- Could you provide geometric intuition or empirical visualization showing how hyperbolic curvature mitigates over-squashing compared to Euclidean path aggregation?
- How does the anti-symmetric update interact with curvature learning, do both converge jointly, or could they interfere?
- The inductive results on WN18RR indicate instability (and missing ranking with bold and underline). Did you test curvature regularization or shared curvature across relations to stabilize training?
- Could you quantify the computational overhead of performing Möbius operations versus standard Euclidean ones?
- Which baselines in Table 1 were re-run under your setup, and which were taken from original sources?
Do you expect the reported performance gaps to hold if all methods were trained with identical random seeds and data splits?
- Will you report mean ± std across multiple seeds for all methods (or at least for your model and the strongest baselines) using identical splits and evaluation protocols?


Overall, the paper is technically solid and clearly written, with a well-motivated contribution.
However, for the results to be fully convincing, they must be statistically validated (e.g., by reporting mean ± std over multiple seeds and ensuring fair re-runs of baselines).
I will be happy to increase my score once these points,  especially the statistical validity, but also the fairness and broader comparison aspects, are properly addressed.

---

### Official Review · Reviewer_Tdhv · 2025-10-31

**Soundness:** 4
**Presentation:** 3
**Contribution:** 3
**Rating:** 4
**Confidence:** 4

**Summary:**

The paper proposes HyPNet, a novel approach for link prediction combining hyperbolic embeddings with path aggregation with the aim of obtaining the best of both approaches. The paper builds on existing literature for both approaches, and ultimately proposes a fully hyperbolic path-based aggregation scheme. In the process, the paper shows that this combined approach is more robust against oversmoothing and maintains the performance strengths of path-based approaches in the link prediction context.

On the empirical side, this paper conducts a comprehensive experimental analysis of the work on both transductive and inductive versions of FB15k-237 and WN18RR, showcasing strong performance, and also highlighting weaknesses transparently.

**Strengths:**

- The conceptual arguments underpinning the design of HyPNet are sound and well-motivated
- The model can be useful for the community to build on, and offers interesting perspectives on the intricacies of link prediction.
- The paper offers a refreshingly honest assessment of strengths and weaknesses, openly and transparently showing where the model fails and the challenges it must overcome.
- The paper's ablation studies are carefully designed and showcase the key points of the work effectively.

**Weaknesses:**

- The experimental results of this work are inconsistently strong, as the authors themselves highlight. This in itself is not problematic. However, it does warrant a deeper exploration on other benchmarks to better understand the failure and success modes of this approach. Other baselines like YAGO3-10 could be an interesting use case to better determine circumstances where HyPNet is effective. At the moment, the main suggestion I would offer the authors is to better define when your model is uniquely well placed to perform at SOTA level. There are some abstract / high-level indications provided vis-a-vis FB15k-237 and WN18RR, but these can be enriched with further experiments.

- The authors highlight problems training this model and reliably obtaining strong performance, e.g., with overfitting and validation / test differences. This is potentially a major bottleneck for the usability of the approach, so I would ask that the authors elaborate more on how this issue presents itself and to conduct more detailed analyses on this point.

Overall, I see merit in this work, and believe it offers added value to the link prediction literature. I also admire its honest and objective assessment of its own contributions. That said, the work would benefit from more experimental analysis to better clarify its strengths and weaknesses, and needs to more strongly showcase its usability in broader contexts. As a result, I very marginally lean against supporting this paper at this stage, but am very open to raising my score should my concerns be addressed.

**Questions:**

N/A

---

### Official Review · Reviewer_hr8D · 2025-10-31

**Soundness:** 1
**Presentation:** 1
**Contribution:** 1
**Rating:** 2
**Confidence:** 4

**Summary:**

The paper aims to address the limitations of graph neural networks (GNNs) for link prediction for knowledge graphs (KGs). The authors argue that prior methods that use hyperbolic space or path aggregation possess certain advantages (respectively, in modeling hierarchies and alleviating over-smoothing), but they still suffer from over-squashing. Their proposal (HyPNet) combines hyperbolic path aggregation with curvature stabilization and an antisymmetric weight update to mitigate over-squashing.

**Strengths:**

- The over-squashing problem is studied in the context of simple graphs, but to the best of my knowledge this is the first study on knowledge graphs.
- The experimental results indicate some incremental improvements across the board.

**Weaknesses:**

- There is little justification for the design choices. The authors argue that hyperbolic and path-based approaches are good for certain reasons, but they suffer from over-squashing -- and then they combine these ideas with learnable curvature. The significance of their theoretical analysis on convergence is unclear and it is also straightforward.

- There are many unsubstantiated claims in the paper, i.e.: "Since the path aggregation conditioned on the query alleviates the over-smoothing problem, path-based methods demonstrate superior performance and versatility." The reason why models such as NBFNets perform well is hidden in their theoretical expressivity and this is well-understood (see; Huang et al, A Theory of Link Prediction via Relational Weisfeiler-Leman on Knowledge Graphs, NeurIPS 2023). In fact, authors show in their own Fig 2 that NBFNets also suffer from over-smoothing.

- The method is based on Bellman Ford Curvature which was used by Topping et al (2022). It appears very incremental and actually transfers all of its weaknesses -- including inflexibility.

- As a more general comment, this kind of rewiring/curvature methods sometimes improve on benchmarks, because it becomes somewhat "easier" to fit the data, but I do not think these methods necessarily generalize well. This can already be seen to some extent in the inductive link prediction setup and the training instabilities -- and it will likely manifest more strongly in out-of-distribution settings. Relatedly, it is unclear whether this approach can be used for building knowledge graph foundation models (see; e.g., Galkin et al, ULTRA: Towards Foundation Models for Knowledge Graph Reasoning, NeurIPS 2024) which perform much better than the reported results in this paper.

**Questions:**

- Is there a way to overcome training instabilities? Can you provide more justification on why this method should be used instead of models such as NBFNets which are generally very stable? Can this method be used in a setup similar to knowledge graph foundation models?

---

### Note · Authors · 2025-11-27

I have read and agree with the venue's withdrawal policy on behalf of myself and my co-authors.